# COVID-19 Vaccines against Omicron Variant: Real-World Data on Effectiveness

**DOI:** 10.3390/v14102086

**Published:** 2022-09-20

**Authors:** Yousra Kherabi, Odile Launay, Liem Binh Luong Nguyen

**Affiliations:** 1Assistance Publique Hôpitaux de Paris, Hôpital Cochin, Centre d’Investigation Clinique, 75014 Paris, France; 2Inserm, CIC 1417, F-CRIN, I-REIVAC, 75014 Paris, France; 3Faculté de Santé, Université Paris Cité, 75006 Paris, France

**Keywords:** COVID-19 vaccines, SARS-CoV-2, omicron, effectiveness

## Abstract

The efficacy of vaccines against coronavirus disease 2019 (COVID-19) has now been well established in phase III clinical trials. However, clinical studies based on real-world data remain critical to assess vaccines effectiveness (VE), especially in specific populations and against variants of concern (VOC). This review presents the principles and methods of VE studies and the main available results on VE of COVID-19 vaccines at the time of Omicron circulation. References for this narrative review were identified through searches of PubMed database up to 13 September 2022. The results of phase III clinical trials have been globally confirmed by VE in real-life studies, including in the elderly. Emergence of VOC Omicron emphasized the importance of booster doses to maintain a high level of protection against severe forms. There are still numerous challenges regarding booster(s) and duration of immunity, particularly in specific subpopulations, and regarding the need for adapted vaccines.

## 1. Introduction

The emergence of SARS-CoV-2 variants of concern (VOC) has illustrated the value of real-world data on vaccine effectiveness (VE). Many pending questions remain after the phase III studies efficacy results: which effectiveness in severe forms, transmission, duration of protection, effectiveness in vulnerable populations such as immunocompromised patients and elderly, and safety data at a larger scale [1]. Updated information from real-world data (or phase IV) led to comprehensive re-evaluations of vaccination recommendations.

Real-world data came from either medical sources (medical records or health departments) or registries (epidemiological surveillance systems or medico-administrative databases). Medical sources provide specific information (such as the presence of comorbidities) but are generally limited in sample size. These types of data have allowed the first detailed descriptions of vaccine failures or serious side effects [2,3,4]. On the other hand, databases have a very large sample size, allowing higher statistical power, but they are less detailed and subject to numerous biases. Coronavirus disease 2019 (COVID-19) pandemic response created and mobilized numerous data from various registries, thus VE data had been obtained promptly [5]. These data are generally analyzed using two methods: case-control studies and cohort studies (Table 1).

Case-control studies are a very interesting way to assess VE because they are easier to perform and are less expensive than cohort studies [6]. Controls should have the same exposure prevalence as the source population from which the cases come from [7]. When cases have one specific infection (e.g., SARS-CoV-2 infection) diagnosed in a medical setting, controls are traditionally selected either from the general population (“healthy controls”), or from among patients with diseases that are not known to be associated with both the infection (SARS-CoV-2 infection) and exposure of interest (COVID-19 vaccines).

Case-control studies also allow test-negative design analyses that consist of comparing the proportion of vaccinated persons among patients infected by SARS-CoV-2 to the proportion of vaccinated persons among patients not infected with SARS-CoV-2 [8]. In contrast to the classical case-control design, controls and cases in the test-negative design have the same clinical presentation: controls are only distinguished from cases by negative laboratory results (Figure 1). An important advantage of test-negative design compared to traditional case-control studies is the relevance of enrolling cases and controls with the same clinical case definition, in the same medical setting, thus guaranteeing that they both have arisen from the same source population and thereby reducing potential selection biases [9]. This methodology is the favored method to assess VE because it makes the measurement of VE possible in the real-world and/or in severe forms of the disease [5,10].

Cohort studies, however, allow longitudinal analyses, such as survival analyses, and thus offer a dynamic overview of VE by measuring the incidence of infection [11]. They were set in the beginning of the COVID-19 pandemics and provided precious information regarding the protection of previous infections or VE (SIREN cohort). However, their usefulness with Omicron VOC is more limited. Some of them were not maintained, and vaccine coverage and vaccine schedule updated quickly, as most of the cohorts were from healthcare workers.

The objective of this review is to present the most recent findings in VE of COVID-19 vaccines, in the context of Omicron circulation, and to recall the general methodological principles of real-world studies in vaccination.

## 2. Materials and Methods

Electronic searches for studies were conducted using PubMed until September 13, 2022, using the search terms “coronavirus”, “SARS-CoV-2”, “Omicron”, “COVID-19”, “effectiveness”, “neutralization assays”, “neutralization antibodies”, in addition to the scientific or commercial names of the vaccines that have been authorized in at least one country. We only selected original studies that reported COVID-19 VE using real-world data at the time of Omicron circulation. Of note, in this narrative review, we did not exclude studies that were not cohort studies, case-control studies, or test-negative design studies. Regarding seroneutralization assays, we extracted the dosage and fold decrease in geometric mean titer for 50% neutralization compared to the SARS-CoV-2 Wuhan reference strain and/or other strains as specified.

## 3. Results

### 3.1. Omicron VOC and General Population Data

#### 3.1.1. What Do We Know about Omicron VOC?

The World Health Organization (WHO) defines a VOC for SARS-CoV-2 if it shows, through comparative evaluation, association with at least one of the following three changes: increased transmissibility; increased virulence or altered clinical presentation; decreased effectiveness of public health and social measures or available diagnostic tools, vaccines, and treatments [12].

In settings where public health strategies rely heavily on high vaccine coverage, the emergence of new variants is bound to have an impact on VE. The Omicron VOC (or B.1.1.529, according to the Pango lineage nomenclature) has more than one hundred sublineages, among which the main ones are: BA.1 (the original sublineage), BA.2, BA.3., BA.4, and BA.5 [12,13].

BA.1 was first designated as an Omicron variant of concern on 26 November 2021, and was initially described as a variant more likely than Delta to target the upper respiratory tract—where vaccination may not prevent infection [14]—whereas the Delta variant infected the lower respiratory tract more [15]. Indeed, the first Omicron variants have shown lower fusogenicity in vitro and a decreased lung infectivity—compared to Delta and the ancestral SARS-CoV-2 strain—in animal models [16]. However, human in vivo infectivity and pathogenicity is difficult to compare, as Omicron spread in a previously vaccinated population. BA.2 had a growth advantage over BA.1 and rapidly replaced it as the main circulating variant at the beginning of 2022 [17,18]. BA.3 was characterized by limited transmissibility, showing that some sublineages can randomly have an uncompetitive capacity in spreading [19]. BA.4 and BA.5 were first detected, respectively, in January and February, 2022, in South Africa, where they have rapidly become the main circulating variants. Compared to BA.1 and BA.2 sublineages, BA.4 and BA.5 both harbor amino-acid substitutions in the spike receptor binding domain, suggesting a significant change in antigenic properties. Omicron sublineages BA.4 and BA.5 have indeed already shown higher transmissibility than the BA.2 lineage [20]. BA.5 had become the main variant in Portugal at the end of May 2022, accompanied by an increase in COVID-19 case numbers and test positivity rate [21,22,23]. At the time of writing, BA.4 and BA.5 have become dominant in Europe [24].

Omicron sublineages are particularly characterized by their capability to evade immunity in convalescent COVID-19 patients and fully-vaccinated individuals [25]. BA.1 and BA.2 have been proven to indeed reinfect convalescent COVID-19 patients [26,27]. Initial immunogenicity studies have confirmed a virological escape of Omicron with a lower neutralizing antibody titer, especially with two doses, but with positive data on the effect of a booster (mRNA vaccines) [28,29]. Moreover, BA.4 and BA.5 harbor the L452Q/R mutation that could enable humoral immunity escape [20]. These sublineages seem to be able to escape from the immunity induced by BA.1 or BA.2 infections [20]. Using the serum from BA.1 breakthrough infections, there seems to be significant reductions in the neutralization of BA.4/5, suggesting the possibility of several Omicron infections in the same individual [13].

#### 3.1.2. Vaccine Effectiveness against Omicron Variant

In epidemiological studies, assessing VE goes with methodological difficulties linked to many potential biases that need to be taken into account when interpreting the results: comparability between vaccinated and unvaccinated populations (especially in a context of a health pass), and previous SARS-CoV-2 infections providing a varying degree of protection depending on time of infection and variants [30,31].

It has been demonstrated that primary series vaccination failed to prevent the outbreak of Omicron variants [32,33,34,35,36,37]. In vitro, neutralizing antibodies in sera from individuals who have received a 2-dose primary series vaccination with Ad26.COV2.S (Johnson & Johnson) were not efficient against BA.1 [27]. In BNT162b2 (Pfizer/BioNTech) sera recipients, serum neutralizing ability against BA.1 was decreased 35-fold compared to prototypical D614G variant, even being ineffective in some sera samples [35,36,38]. BA.2 and BA.3 also showed a particularly important vaccine immune escape [39]. BA.4/5 have also showed reduced neutralization by the serum from individuals vaccinated with triple doses of ChAdOx1 (Oxford/AstraZeneca) or BNT162b2 vaccine compared to BA.1 and BA.2 sublineages [40]. Compared to primary series vaccination, booster doses of vaccines can increase serum neutralizing titers against Omicron sublineages [32,35,36]. BNT162b2 booster dose significantly improved neutralization efficiency against BA.1 [41].

We summarized the data from the main studies, assessing primary series vaccination VE against Omicron in Table 2. In most cases, Omicron infection in vaccinated individuals was mild. The first available data from South Africa, the United Kingdom, and Qatar were reassuring about the protection from severe forms of the disease. A first study in South Africa from an insurance company found a protection rate of 70% for hospitalized forms with two doses, but 33% for all clinical forms combined [42]. However, the article did not report previous history of infection and the time between the second dose and the events. Additional information came from the vaccine surveillances reports published by the UK Health Security agency and from a British study which measured VE against Omicron compared to the Delta variant, confirming a low VE on symptomatic forms but a relatively preserved effectiveness on severe forms, confirmed by a test negative design study among healthcare workers in the USA [37,43,44]. More recently, a Qatari study confirmed the low VE on symptomatic forms which decreases rapidly after 2 doses, but also after 3 doses. Even so, the effectiveness against severe forms remains good, at 76.5% with a booster [45]. Moreover, there are emerging data on humoral immunity around Omicron breakthrough infections [2,3]. Data on Omicron breakthrough infections are limited to small cases series at the time of writing, in patients receiving two or three doses [46,47]. However, further data is needed on vaccine-induced cellular immunity, which would play an important role in protecting against severe forms and would be less sensitive to mutations of the variants [48]. Vaccine-induced cellular immunity seems to indeed allow a broader protection compared to humoral immunity, including against Omicron [49,50].

To curb the decline in effectiveness after the third dose, a fourth dose has been proposed, particularly for the elderly, for whom previous studies have shown a greater decline in VE over time [55].

A non-randomized clinical Israeli study assessing the immunogenicity and safety of a fourth dose of a mRNA vaccine—either BNT162b2 or mRNA-1273 (Moderna)—in 274 healthcare workers, 4 months after the last injection of a series of three BNT162b2 doses, was recently published [56]. In this study, 25.0% of the participants in the three-doses group were infected with Omicron, as compared with 18.3% of the participants with a fourth dose of BNT162b2 and 20.7% of those with a fourth dose of mRNA-1273. VE against any SARS-CoV-2 infection was 30% (95% Confidence Interval (CI), −9 to 55) for BNT162b2 and 11% (95% CI, −43 to 44) for mRNA-1273. In order to corroborate these VE findings, immunological response was also assessed. A fourth dose yielded similar humoral response and particularly similar levels of Omicron-specific neutralizing antibodies compared to the peak response one month after the third dose. These results suggest that the best humoral immunogenicity possible of mRNA vaccines is achieved after three doses and that antibody titers can be restored by a fourth dose [56]. In a larger multicentric study including 29,611 healthcare workers, the breakthrough infection rate among those who received a fourth dose of BNT162b2 was 6.9% compared with 19.8% in those who received three doses (relative risk, 0.35; 95% CI [0.32–0.39]) [56]. In another Israeli study, including participants who were 60 years of age or older and who had received three doses of BNT162b2, the adjusted rate of proven SARS-CoV-2 infection in the fourth week after administration of the fourth dose was lower than that in the three-dose group by a factor of 2.0 (95% CI [1.9–2.1]) [57]. These findings were substantiated by another nationwide study in Israel in which the difference, when comparing three doses with four doses, was 180.1 cases per 100,000 persons (absolute risk, 95% CI [142.8–211.9]) for COVID-19-related hospitalization and 68.8 cases per 100,000 persons (absolute risk, 95% CI [48.5–91.9]) for severe COVID-19 [58]. A Canadian test negative design study, including 13,654 long term care residents who tested positive for Omicron infection and 205,862 test negative controls, has been recently published [59]. In this study, compared with a third dose of mRNA COVID-19 vaccine, a fourth dose improved protection against infection (marginal effectiveness, 19% 95% CI [12–26]), symptomatic infection (31% [20–41]), and severe outcomes (40% [24–52]).

On 11 July 2022, the European Centre for Disease Prevention and Control (ECDC) and the European Medicines Agency COVID-19 task force (ETF) both agreed to recommend a second booster of mRNA COVID-19 vaccines in adults aged 60 years and over and ‘people with medical conditions putting them at high risk of severe disease’. At the time of writing, it is considered that there is no strong evidence to support the current use of a second booster in immunocompetent persons below 60 years old [60].

### 3.2. Vulnerable Populations

Knowledge of VE in immunocompromised individuals and the elderly is particularly important, as they are at risk of having a lower vaccine response but also of developing severe forms. Moreover, these vulnerable populations have been excluded from clinical trials. The high effectiveness of mRNA and viral vector vaccines in the elderly, and particularly of the booster dose, has raised expectations of good VE in immunocompromised patients [61]. Given the diversity in the type and level of immunosuppression, as well as very high vaccine coverage in this population, epidemiologic studies are difficult to conduct and to interpret. Including a sufficient sample for each vulnerable population is indeed a challenge, and most studies cannot achieve statistical significance on their own. Immunogenicity studies mostly retrieved measurements of antibody titer, but only a few studies assessed the rate of neutralizing antibodies. Thus, interpretation in case of a positive result is limited by the heterogeneity of measurement techniques and the absence of a correlate of protection. However, several literature reviews and meta-analyses agree that there are no safety concerns, but that immunocompromised individuals respond less well overall to vaccination [62].

While the data are rather reassuring for patients on dialysis or with solid cancer, patients with hematological diseases, solid organ transplants, or who are treated with lymphopenic agents seemed to be at higher risk of infection, symptomatic disease, and severe illness, despite primary series vaccination [63,64,65,66]. Several studies have shown an augmentation of humoral and cellular immune responses after a third dose of SARS-CoV-2 vaccination and viral neutralization in patients with multiple myeloma, but these responses seemed to be decreased against the Omicron variant [67,68]. A French observational study has shown that, among 25 patients with lymphoid malignancies and positive anti-spike titers before the third dose, 92% patients increase their anti-spike and neutralizing antibody titers after the boost. All 18 initially seronegative patients remained negative.

Severe cases of COVID-19 and poor immune response after two doses of vaccine have been described in patients with solid organ transplants (SOT) [69,70]. Most studies assess the immunogenicity of the COVID-19 vaccine, despite the absence of a correlate of protection. As described for immunocompetent persons in previous studies, administration of a third dose of the BNT162b2 vaccine to SOT recipients significantly improved the immunogenicity of the vaccine [71]. These immunogenicity data were correlated with no cases of COVID-19 reported in any of the patients in a study including 101 SOT recipients before Omicron emergence. In a study including 53 patients with SOT, a third dose improved antibody responses against all SARS-CoV-2 variants except Omicron, where antibody responses and neutralizing activity remained suboptimal [72]. In another study which included 395 SOT recipients receiving a third dose of the BNT162b2, focusing on the immune response, the increased antibody titer was significantly higher among patients with detectable antibodies after the second dose than those without [73]. Overall, 22.1% of participants did not develop any humoral response. Not surprisingly, cumulative time from transplantation and liver recipients were both positively associated with the presence of a humoral response, whereas older age, administration of prednisolone, and proliferation inhibitors were associated with lower antibody titers. This study may suggest, although no correlate of protection has been established yet, that patients at risk of a suboptimal immune vaccine response maybe require repeated booster doses and/or alternative treatment approaches.

COVID-19 VE against Omicron infection and hospitalization in patients taking immunosuppressive medications have been assessed in a retrospective cohort study led in the United-States, including 5609 patients receiving immunosuppressants [74]. In this population, three doses of BNT162b2 had a VE of 50% (95% CI [31–64]; *p* < 0.0001) and three doses of mRNA-1273 had a VE of 60% (95% CI [42–73]; *p* < 0.0001) against Omicron infection. Three doses of either vaccine had a VE of 87% (95% CI [73–93]; *p* < 0.0001) against hospitalization due to COVID-19. Being treated by conventional synthetic disease-modifying antirheumatic drugs (DMARDs) (hazard ratio (HR) 2.32, 95% CI [1.23–4.38]; *p* = 0.0097) or glucocorticoids (HR 2.93, 95% CI [1.77–4.86]; *p* < 0.0001) and having a history of SOT or bone marrow transplantation (HR 3.52, 95% CI [2.01–6.16]; *p* < 0.0001) were associated with increased risk of COVID-19-related hospitalization compared to immunocompetent controls.

## 4. Discussion

Vaccination is one of the most effective ways to control SARS-CoV-2. This narrative review highlights the fact that VE against Omicron is improved by boosters but wanes over time. We still do not know how long immunity lasts after the fourth dose. The emergence of VOC triggers the question of the number of boosters in the general population, as the disease is very likely to become endemic. We still do not know how vaccine needs will change and if we will need to administer a booster every time immunity drops.

Indeed, more than the level of VE, its duration seems to be the most important question. In a systematic review assessing the duration of effectiveness of vaccination against COVID-19 caused by Omicron, Higdon and colleagues have shown that VE against severe COVID-19 was lower than that observed in the pre-Omicron period. Boosters improved VE against Omicron, which remained high four months after vaccination. VE against symptomatic disease decreased faster for Omicron than previous VOC, with fading protection by 4–6 months. Of note, protection after booster vaccination seemed to decrease quickly, although less than after third doses [75].

Recently, a “Omicron mRNA-vaccine” seemed to show higher efficacy against BA.1 infection than the traditional mRNA vaccine [36]. In a press release (data not peer-reviewed), Pfizer and BioNTech have presented the preliminary results of two Omicron-adapted COVID-19 vaccine candidates (one monovalent and the other bivalent) as a combination of the Pfizer-BioNTech COVID-19 vaccine and a vaccine candidate targeting BA.1 spike protein. A booster dose of both Omicron-adapted vaccine candidates seems to have elicited a higher immune response against Omicron BA.1 as compared to BNT162b2. It is thus not certain if these vaccine candidates will be effective against BA.4/5 sublineages or potential future variants [76]. By 1 September 2022, the first bivalent COVID-19 booster vaccine was approved by the Food and Drug Administration (FDA), the European Medicine Agency (EMA), and by the UK medicines regulator [77,78].

Pfizer-BioNTech recently presented an adapted bivalent vaccine, targeting the Omicron subvariants BA.4 and BA.5 in addition to the original strain of SARS-CoV-2. At the time of writing, data on its efficacy have not been published yet, however the EMA has recommended its authorization by 12 September 2022 [79].

The booster question also raises ethical issues, as low-resources countries still lack vaccines for their population. Booster doses are offered in various regions such as Israel, the European Union, and the USA, but at least a billion individuals in Africa are still unvaccinated. At the time of writing, only 21% of people in low-income countries have received at least one dose of vaccine [80]. Unvaccinated people are more likely to allow viral spreading, and transmission of the SARS-CoV-2 in these settings is a major factor for the emergence of new VOC that could be more transmissible and/or better escape immunity. Although SARS-CoV-2 infection is expected to become endemic, there is still a risk of the emergence of new lineages and variants, the global consequences of which are difficult to predict [81].

## 5. Conclusions

The COVID-19 pandemic has had a considerable impact in terms of public health, psychology, and economy, and these effects have yet to be measured. However, it is clear that it has also allowed the development and large-scale use of a new generation of vaccines (mRNA and viral vectors) in record time. The acceleration of clinical research, with the promising results of clinical trials, has been globally confirmed by real-life studies. However, these studies provide crucial information on rare and serious adverse events, as well as on effectiveness in specific populations. Many questions still need to be answered in order to best adapt vaccine policies against this disease, which is expected to become endemic.

## Figures and Tables

**Figure 1 viruses-14-02086-f001:**
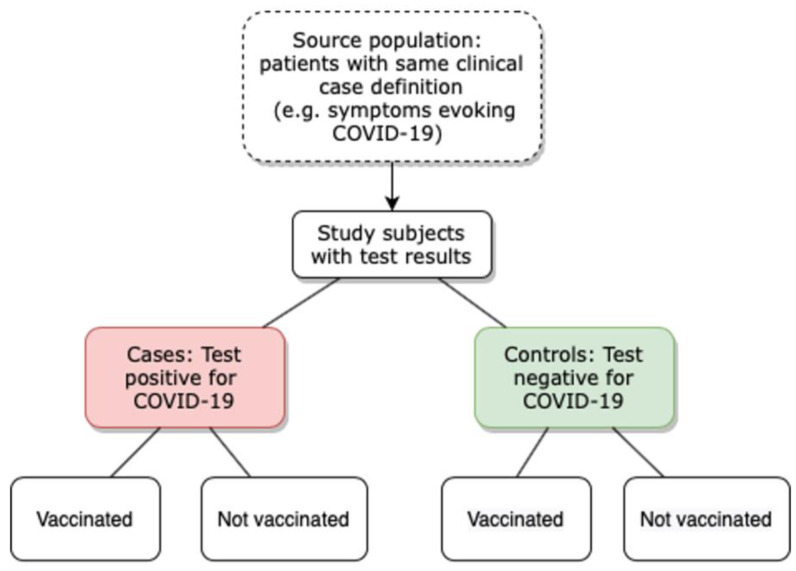
Principles of test-negative design in COVID-19 vaccine studies.

**Table 1 viruses-14-02086-t001:** Summary of methods for measuring real-life effectiveness in the context of COVID-19 vaccination.

	Cohort Studies	Case-Control Studies
Description	Longitudinal follow-up of two populations according to the exposure of interest (vaccination)	Selection of subjects presenting the event of interest (SARS-CoV-2 infection)
Main analysis (main outcome)	Comparison of incidence (Hazard Ratio)	Comparison of prevalence (Odds-Ratio)
Assets	Detailed prospective data Follow-up over time Allow “nested” case-control studies	Fast to analyze Affordable Suitable for large databases Suitable for rare events (COVID-19 hospitalized)
Limits	Expensive: limited participants Not suitable for rare events	Retrospective data Not suitable for rare exposure (high vaccine coverage COVID-19) Many biases (selection, reporting)

**Table 2 viruses-14-02086-t002:** Main studies assessing real-life vaccine effectiveness against Omicron.

Reference	Country	Study Design	Study Population	Vaccine Assessed (Both Homologous and/or Heterologous)	Omicron Infection Rate in People Vaccinated with 3 Doses	Omicron Infection Rate in People Vaccinated with 2 Doses	Omicron Infection Rate in Unvaccinated People
Accorsi et al. 2022 [51]	USA	Test negative design	Adults ≥ 18 years old with with COVID-like illness and PCR results available	BNT162b2 mRNA-1273	2441/21,028 * (11.6%)	7245/26,701 * (27.1%)	3412/12,133 * (28.1%)
Acuti Martelluci et al. 2022 [52]	Italy	Cohort	Adults and children with no positive SARS-CoV-2 swab at cohort beginning	BNT162b2 mRNA-1273 ChAdOx1 Ad26.COV2.S	15,927/250,354 (6.4%)	79,787/672,653 (11.9%)	41,281/293,702 (14.1%)
Andrews et al. 2022 [37]	UK	Test negative design	Adults ≥ 18 years old with COVID-like illness and PCR results available	BNT162b2 mRNA-1273 ChAdOx1	753,437/2,038,969 (37%)		101,109/244,716 (41.3%)
Collie et al. 2022 [42]	South Africa	Test negative design	Adults ≥ 18 with PCR results during Omicron circulating period	BNT162b2 ^£^	NR	9700/45,657 (21.2%)	7889/26,331 (30.0%)
Ferdinands et al. 2022 [53]	USA	Test negative design	Adults ≥ 18 years old with COVID-like illness and PCR results available	BNT162b2 mRNA-1273	1938/10,931 (17.7%)	8351/19,822 (42.1%)	13,991/24,799 (56.4%)
Kirsebom et al. 2022 [54]	UK	Test negative design	Adults ≥ 18 years old with COVID-like illness and PCR results available	BNT162b2 mRNA-1273 ChAdOx1	437,276/999,124 (43.8%) *		59,793/97,073 (61.6%) *

Abbreviations: USA, United States of America; UK, United Kingdom; NR; not reported. * This study retrieved symptomatic infections only. Asymptomatic infections were not analyzed. ^£^ This study assessed only homologous vaccination with BNT162b.

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
