# Peer review of "COVID-19 Vaccines against Omicron Variant: Real-World Data on Effectiveness"

_viruses, 2022, doi:10.3390/v14102086_

Round 1

Reviewer 1 Report

This manuscript tried to review post-licensure studies of vaccine efficacy against Omicron variants. The literatures were collected from PubMed and Google Scholar databases using the search terms: “SARS-CoV-2”, “Omicron”, “COVID-19”, “effectiveness”, “neutralization assays”, “neutralization antibodies”

Firstly, I am concerned about the search strategy the authors employ. The terms used for example, certainly would not include keyword ‘coronavirus’. Moreover, the search strategy is pretty basic, not sure if the study is qualified enough, especially when a grey database such as Google Scholar is used. Methods of data extraction are also important to be highlighted.

Secondly, I am concerned on the type of studies and patients’ characteristics which were not properly discussed. Narrative explanation made by the authors does not allow readers to be capable of identifying the foregoing information. Hence, it is very difficult to make a solid conclusion. Also, other factors affecting the efficacy should be discussed too.

For improvement, I suggest reevaluating your search strategy. Then, you could make better presentation by tabulating the results. The narrative explanation is good, but it’s very hard to follow.

The format seems like a systematic review, but the content is more like a narrative review. Please reconsider the format of the manuscript.   

Author Response

Reviewer 1

This manuscript tried to review post-licensure studies of vaccine efficacy against Omicron variants. The literatures were collected from PubMed and Google Scholar databases using the search terms: “SARS-CoV-2”, “Omicron”, “COVID-19”, “effectiveness”, “neutralization assays”, “neutralization antibodies”

Firstly, I am concerned about the search strategy the authors employ. The terms used for example, certainly would not include keyword ‘coronavirus’. Moreover, the search strategy is pretty basic, not sure if the study is qualified enough, especially when a grey database such as Google Scholar is used. Methods of data extraction are also important to be highlighted.

Answer: Our work in a narrative review as stated in the Abstract, the Introduction and in the Methods section. We revised our methodology which is now qualified enough to satisfy the requirements for a narrative review (lines 86-93). As suggested, we excluded Google Scholar and we added ‘coronavirus’ as a key word in our search. Methods of data extraction were also explicited in the Methods section.

Secondly, I am concerned on the type of studies and patients’ characteristics which were not properly discussed. Narrative explanation made by the authors does not allow readers to be capable of identifying the foregoing information. Hence, it is very difficult to make a solid conclusion. Also, other factors affecting the efficacy should be discussed too. For improvement, I suggest reevaluating your search strategy. Then, you could make better presentation by tabulating the results. The narrative explanation is good, but it’s very hard to follow. 

Answer: We thank the reviewer for his/her suggestion. We reframed the Results section and added a Table to summarize the main findings (Table 2).

The format seems like a systematic review, but the content is more like a narrative review. Please reconsider the format of the manuscript. 

Answer: We thank the reviewer for his/her comment. As explained above, the work presented here is a narrative review. We insisted on that point in the Methodology section and the whole format of the manuscript is compatible with the traditional frame of a narrative review.

Reviewer 2 Report

This is an interesting review of vaccine effectiveness (VE) data that has been published since the completion of the phase 3 efficacy trials and widespread use of COVID19 vaccines. The overall question of how best look at real world evidence for VE is an important one and organization of the review good. The authors. There are no real novel conclusions however which limits the impact of the work.

Major Comments:

1) In the table of Study Design, Assets section, the case control studies lists that less participants are needed. I'm not sure this is a true statement and seems contradictory to the rest of the table.

2) There is a lot of discussion in the Intro comparing Cohort studies to Case-Control Studies but the vast majority of the article describes the latter, not the former. I think this entire section should be revised to just discuss the fact that most post-marketing studies at Case-Control design and how VE was in general described in the studies that will be discussed in the rest of the manuscript as well as why VE studies are important.

3) There should be better description of the results of the electronic searches and how articles were selected.

4) The results sections would be clearer with some tables which illustrates VE findings for various studies (primary versus booster studies, 3rd vs 4th dose, omicron versus other variants etc)

5) Page 4, line 106 is a misleading statement. I don't  believe there were more URTI with Omicron compared to Delta infections and the authors should comment on the fact that there were more vaccinated persons during the omicron wave.

6) Page 5, Line 164: It's not clear what is meant by this sentence.

7) On page 5, second paragraph, the authors sometimes confuses decreases in neutralizing AB titers with VE data. Please review. For example, reference 48.

8) A better way of describing the limited data in immunocompromised host would be to understand how the responses compare to those who are immunocompetent rather than just reporting the findings.

Minor Points
1) Would refer to the "routine dose" as the primary series

Author Response

Reviewer 2

This is an interesting review of vaccine effectiveness (VE) data that has been published since the completion of the phase 3 efficacy trials and widespread use of COVID19 vaccines. The overall question of how best look at real world evidence for VE is an important one and organization of the review good. The authors. There are no real novel conclusions however which limits the impact of the work. 

Major Comments:

  • In the table of Study Design, Assets section, the case control studies lists that less participants are needed. I'm not sure this is a true statement and seems contradictory to the rest of the table.

Answer: Case control studies generally requires few study subjects because the cases are identified at study onset and the outcomes have already occurred with no need for a long-term follow-up. In a cohort study, there is a need to recruit much more participants in order to eventually obtain the same number of cases. This is further elaborated in an article published by Mann et al. N in the Emergency medicine journal (doi:10.1136/emj.20.1.54) and by Irony in the latest JAMA Guide to Statistics and Methods (doi:10.1001/jama.2018.12115).

2) There is a lot of discussion in the Intro comparing Cohort studies to Case-Control Studies but the vast majority of the article describes the latter, not the former. I think this entire section should be revised to just discuss the fact that most post-marketing studies at Case-Control design and how VE was in general described in the studies that will be discussed in the rest of the manuscript as well as why VE studies are important.

Answer: We thank the reviewer for his/her suggestion. We agreed with the reviewer on the fact that case-control studies and more particularly test negative designs represent the vast majority of VE studies. While cohort studies have been used in the beginning of the pandemic to assess VE, they are very costly to maintain, and the high vaccine coverage among the participants made the calculation of VE very complex. We reviewed the Introduction section to make it clearer (lines 64-71).

  • There should be better description of the results of the electronic searches and how articles were selected.

Answer: We agree with the reviewer, which comment has also been underlined by Reviewer 1. We modified this section, keeping in mind that we are here reporting a narrative review and not a systematic review.

  • The results sections would be clearer with some tables which illustrates VE findings for various studies (primary versus booster studies, 3rd vs 4th dose, omicron versus other variants etc)

Answer: We thank the reviewer for his/her suggestion. As also suggested by Reviewer 1, we added a Table 2illustrating findings of the main studies.

  • Page 4, line 106 is a misleading statement. I don't  believe there were more URTI with Omicron compared to Delta infections and the authors should comment on the fact that there were more vaccinated persons during the omicron wave.

Answer: We thank the reviewer for his/her suggestion. We agree with the reviewer that this statement is not clear. We explicited this part to make it more clear for the readers (lines 112-117).

  • Page 5, Line 164: It's not clear what is meant by this sentence.

Answer: We thank the reviewer for his/her suggestion. We changed this sentence to make it more understandable for the reader (lines 177-178).

  • On page 5, second paragraph, the authors sometimes confuses decreases in neutralizing AB titers with VE data. Please review. For example, reference 48.

Answer: We reframed this paragraph to make it clearer for the readers (lines 201-206). We also reviewed the manuscript to correct potential misleading paragraphs.

8) A better way of describing the limited data in immunocompromised host would be to understand how the responses compare to those who are immunocompetent rather than just reporting the findings.

Answer: We agree with the reviewer and made this part more explicit for the readers (lines 278-288).

Minor Points

1) Would refer to the "routine dose" as the primary series

Answer: We thank the reviewer for his/her suggestion. We changed “routine dose” for “primary series vaccination” (lines 150, 152, 159, 163 etc.).

Reviewer 3 Report

Thank you very much to the editor of VIRUSES for allowing me to review the document titled "COVID-19 Vaccines Against Omicron Variant: Real World Data on Efficacy"

The authors have submitted was an interesting review on the state of art of the most recent findings in vaccine effectiveness against COVID-19, considering the current circulation of the Omicron variant.The paper is based on rich and new literature, it is well structured, the figures and tables are easy to understand for the reader and deserves publication. Some comments to be addressed by the authors:

MAJOR:

It is my opinion that the discussion section is poor compared to the topics discussed. There should be a better discussion for example why the current vaccines are less effective against the Omicron variant.

MINOR:

1. line 147: please replace ; with ,

2. For better reading in Section 3.1.2, please indicate the manufacturers of each vaccine mentioned

3. line 142: "In BNT162b2 sera recipients, serum neutralizing ability against BA.1 was decreased 40-fold": the reduction in antibody titer against which variant of SARS-CoV-2 was calculated? compared with the wild-type strain? Please argue further.

Author Response

Reviewer 3

The authors have submitted was an interesting review on the state of art of the most recent findings in vaccine effectiveness against COVID-19, considering the current circulation of the Omicron variant. The paper is based on rich and new literature, it is well structured, the figures and tables are easy to understand for the reader and deserves publication. Some comments to be addressed by the authors:

MAJOR:

It is my opinion that the discussion section is poor compared to the topics discussed. There should be a better discussion for example why the current vaccines are less effective against the Omicron variant.

Answer: We thank the reviewer for his/her thoughtful review of our work and kind words. We agree with the reviewer and completely revised the discussion section.

MINOR:

  1. line 147: please replace ; with ,

Answer: We thank the reviewer for his/her correction. We corrected this mistake (line 159).

  1. For better reading in Section 3.1.2, please indicate the manufacturers of each vaccine mentioned

Answer: We thank the reviewer for his/her suggestion. We added the manufacturers of each vaccine.

  1. line 142: "In BNT162b2 sera recipients, serum neutralizing ability against BA.1 was decreased 40-fold": the reduction in antibody titer against which variant of SARS-CoV-2 was calculated? compared with the wild-type strain? Please argue further.

Answer: We thank the reviewer for his/her correction. The reduction in antibody titer was calculated against a prototypical D614G variant (lines 154-155).

Round 2

Reviewer 1 Report

Please find below my comments:

1.     Since author exclude Google scholar and add a new term, changes in the included studies should be clarified. Note that, I have nothing against the inclusion of Google scholar as the source. But eligibility/selection criteria should be established for such ‘grey’ database.

2.     Please provide more details and use formal report (peer-review one) for the following statement. Otherwise, please revise:

“BA.2 had a growth advantage over BA.1 and rapidly replaced it as the main  circulating variant at the beginning of 2022 [12].”

3.     Kindly make sure once again on the quality of publication included in the review. Informal resources my be included but should be accompanied by a disclaimer that the report is not peer-reviewed or still in a pre-print form for example.

4.     Table. Subject characteristics (or inclusion/exclusion criteria) should be presented. Homologous or heterologous vaccine should be clarified. Also, if the vaccine is heterologous, kindly provide the order of the vaccination. Interval between the first, second, and the third dose should also be disclosed. Make a landscape table if necessary.

5.     Narrative review does not have to follow Introduction, Mtehods, Results, Discussion, format.. Hence, section such as Results and Discussion can be deleted/renamed as necessary.

6.     Recommendation by authors would be much appreciated regarding the vaccine efficacy against Omicron.

7.     Authors are encourage to update their review with the new reported findings that might be published during the review period.

Author Response

We thank the editor and the reviewers for their careful reading of the manuscript and their constructive remarks. We have taken the comments on board to improve and clarify the manuscript. Please find below a detailed point-by-point response to all comments (reviewers’ comments in bold).

Reviewer 1

  1. Since author exclude Google scholar and add a new term, changes in the included studies should be clarified. Note that, I have nothing against the inclusion of Google scholar as the source. But eligibility/selection criteria should be established for such ‘grey’ database.

Answer: We thank the reviewer for his/her comment. Actually, including or excluding Google Scholar as a source and add or withdraw the term “coronavirus” did not change anything in the studies included.

  1. Please provide more details and use formal report (peer-review one) for the following statement. Otherwise, please revise: “BA.2 had a growth advantage over BA.1 and rapidly replaced it as the main circulating variant at the beginning of 2022 [12].”

Answer: We thank the reviewer for his/her comment. We changed the reference to add two formal reports (line 122).

  1. Kindly make sure once again on the quality of publication included in the review. Informal resources my be included but should be accompanied by a disclaimer that the report is not peer-reviewed or still in a pre-print form for example. 

Answer: We thank the reviewer for his/her comment. We agree with the reviewer We did not include any pre-print. We included a press release and thus added a disclamer as suggested (line 313).

  1. Table. Subject characteristics (or inclusion/exclusion criteria) should be presented. Homologous or heterologous vaccine should be clarified. Also, if the vaccine is heterologous, kindly provide the order of the vaccination. Interval between the first, second, and the third dose should also be disclosed. Make a landscape table if necessary.

Answer: We thank the reviewer for his/her comment. We added subject characteristics and clarified homologous/heterologous vaccination as suggested. For the sake of clarity, we preferred to not provide the vaccination order because all the possible combinations are described in each of the cited studies.

  1. Narrative review does not have to follow Introduction, Mtehods, Results, Discussion, format.. Hence, section such as Results and Discussion can be deleted/renamed as necessary.

Answer: We thank the reviewer for his/her suggestion. We kept this format as suggested by the Editor for this special issue.

  1. Recommendation by authors would be much appreciated regarding the vaccine efficacy against Omicron. 

Answer: We thank the reviewer for his/her suggestion. Recommendations are planned to be published in an Editorial accompanying this narrative review.

  1. Authors are encourage to update their review with the new reported findings that might be published during the review period.

Answer: We thank the reviewer for his/her suggestion. We updated our search up to September 13th, 2022.

Reviewer 2 Report

see attached filed

Author Response

We thank Reviewer 2 for his/her careful reading of the manuscript and his/her constructive remarks. Please see the attached file for the point by point response.
